# An integrated model of financial socialization, technology, and financial capability in predicting financial well-being

Nguyen Quoc Anh [ID]*

University of Economics Ho Chi Minh City, Ho Chi Minh City, Vietnam

* quocanh@ueh.edu.vn

## Abstract

This study develops and empirically tests an integrated framework that explains how financial socialisation, technological factors, and financial capability jointly shape financial behaviour and in an emerging economy context. Using data from 306 Vietnamese adults, the study applies Partial Least Squares Structural Equation Modelling to assess direct, mediating, and moderating effects. The results show that both family financial socialization and artificial intelligence significantly enhance financial behaviour and financial well-being, with financial behaviour mediating these relationships. Artificial intelligence exerts a stronger influence on financial behaviour than family financial socialisation, while its impact on financial well-being operates primarily through behavioural pathways. Financial literacy and digital trust significantly strengthen the effect of artificial intelligence on financial behaviour, although the moderating effects are relatively modest. Financial well-being is positioned as the ultimate outcome of the model, and the findings confirm that improvements in well-being are largely driven by behavioural adjustments rather than direct technological exposure alone. The study offers theoretical contributions by integrating social, technological, and capability-based elements into a unified financial well-being framework and highlights the conditional roles of digital trust and financial literacy in shaping AI-driven financial behaviour. It also provides practical implications for financial education and responsible digital finance adoption to enhance financial resilience and long-term well-being.

## 1. Introduction

Enhancing individual and household financial well-being remains a central concern in both academic research and policy discussions [1]. Financial well-being reflects one's ability to manage financial obligations, maintain a desired living standard, and withstand economic shocks [2,3]. It also underpins social inclusion and economic resilience [4]. In the digital era, financial behaviour is increasingly influenced not only

**Data availability statement:** All relevant data are included within the manuscript and its Supporting Information files.

**Funding:** This research is funded (supported) by University of Economics Ho Chi Minh City, Vietnam (UEH). The funders had no role in study design, data collection and analysis, decision to publish, or preparation of the manuscript. Grant No. 2025-12-01-3292 was awarded to Nguyen Quoc Anh.

**Competing interests:** The author has declared that no competing interests exist.

by social learning but also by technological tools, particularly AI-enabled financial systems [5]. This shift calls for an integrated framework that explains how social foundations and digital innovations jointly shape financial behaviour and well-being outcomes [6].

The Family Financial Socialization Theory posits that individuals learn financial values and practices primarily within the family context [7, 8, 9]. Early exposure to constructive financial discussions and parental modeling fosters responsible money management, promoting financial behavior and long-term well-being [10]. However, recent evidence indicates that while family socialisation continues to influence financial well-being, its direct impact on financial behaviour is relatively modest compared to technological factors [11]. Financial behavior, therefore, mediates the relationship between family socialization and financial well-being [9].

Technological innovation, particularly Artificial Intelligence (AI), further transforms financial decision-making. Aligned with SAFE principles, AI enables sustainable finance through automated analytics, personalized investment, and efficient resource allocation [12]. However, limited financial literacy can reduce users' ability to interpret algorithmic recommendations effectively, potentially undermining financial well-being [13]. Empirical findings suggest that AI exerts a stronger influence on financial behaviour than family socialisation, while its effect on financial well-being operates mainly through behavioural pathways [14]. Understanding how AI shapes financial behavior and outcomes remains an emerging research frontier.

Despite growing interest, existing research remains fragmented. Most studies investigate family, technological, or media factors in isolation rather than as interconnected drivers of finance [11]. Few studies simultaneously examine the mediating role of financial behaviour and the moderating roles of financial literacy and digital trust within a unified model, particularly in emerging economies such as Vietnam. These gaps limit theoretical integration and the generalizability of findings.

To address this gap, the study examines family financial socialisation and artificial intelligence together in the same model. It also considers financial literacy and digital trust as conditions that may influence how these factors work. Financial behaviour is treated as the main channel through which these influences translate into financial well-being.

This study makes several key contributions. It integrates the Family Financial Socialization Theory and SAFE principles into a coherent conceptual model that explains how social and technological factors jointly shape financial behaviour and financial well-being. Empirically, the study confirms that financial behaviour serves as a central mediating mechanism linking both family financial socialisation and artificial intelligence to financial well-being. This finding clarifies that improvements in well-being occur primarily through behavioural adjustments rather than through direct technological exposure alone.

The results further demonstrate that artificial intelligence exerts a stronger influence on financial behaviour than family socialisation, highlighting the growing role of digital tools in shaping everyday financial practices. In addition, the study identifies digital trust and financial literacy as significant, though modest, moderating factors,

showing that the behavioural impact of AI depends partly on users' confidence in digital systems and their financial capability. From a policy perspective, the findings suggest that strengthening financial literacy and digital trust is necessary to maximize the behavioural benefits of AI-enabled financial tools. Rather than focusing solely on technology adoption, policymakers and financial institutions should combine digital innovation with capability development and trust-building measures to enhance financial well-being.

The paper proceeds as follows. The next section presents the theoretical foundations and hypotheses. Section three describes the research design, followed by empirical results in section four. The final section discusses implications and future research directions.

## 2. Literature review

### 2.1. Foundation theory

This study is grounded in two theoretical foundations that together explain the development of financial behavior and financial well-being in the modern financial environment (Table 1). The Family Financial Socialization Theory describes how individuals acquire financial knowledge and practical money management skills through early learning and personal financial experiences [7]. These experiences shape budgeting discipline, saving habits, and responsible financial choices, which are central to long-term financial stability and subjective financial well-being. The theory also highlights the role of financial behavior as the mechanism through which foundational financial learning influences well-being outcomes [16]. Although useful, this theory does not fully explain psychological or cognitive processes. It also does not consider the impact of digital technologies, which increasingly shape financial capability.

The SAFE Principles reinforce the importance of sustainability, accountability, fairness, and ethics in technology-enabled financial practices. However, these frameworks do not explain individual differences in technology acceptance, trust in automated systems, or cognitive biases that may affect financial decisions and financial well-being. They also do not address digital capability requirements that influence the effectiveness of Artificial Intelligence [15, 17].

### 2.2. Hypothesis development

The Family Financial Socialization Theory asserts that financial norms, beliefs, and decision-making patterns are shaped early in life through observation, interaction, and practical exposure within the family environment [18]. These formative experiences create cognitive lenses through which individuals interpret financial information, assess risks, and make financial choices throughout adulthood [19]. Financial socialisation is not merely a transfer of knowledge but a process that forms cognitive schemas for money management [20]. Individuals who frequently observe constructive financial

**Table 1. Summary of theories.**

| Theory | Key contribution | Limitations in the financial well-being context | Source |
|---|---|---|---|
| **Family Financial Socialization Theory** | Explains how early financial exposure and personal financial learning shape later financial behavior and financial well-being. Demonstrates that foundational experiences influence budgeting, saving, and responsible money management, which subsequently enhance financial stability and subjective well-being. Clarifies the pathway in which financial behavior mediates the link between early financial learning and financial well-being. | Provides a limited explanation of psychological, cognitive, or motivational mechanisms affecting financial well-being. Does not incorporate behavioral decision-making factors relevant in modern digital financial environments. Needs integration with behavioral, cognitive, or capability-based theories for stronger explanatory power. | [9]. |
| **SAFE Principles** | Show how Artificial Intelligence supports sustainable and informed financial actions, improves planning accuracy, reduces uncertainty, and indirectly enhances financial well-being. | Lack of attention to cognitive biases, digital capability, and digital trust that influence financial well-being. Require integration with technology adoption, digital literacy, and behavioral decision theories to enhance model robustness. | [15] |

practices develop the ability to budget, regulate spending, and maintain discipline. Empirical research shows that stronger financial socialisation corresponds with more rational financial behaviours [9]. This suggests that socialisation serves as the foundational layer upon which financial behaviour is constructed. The theory further suggests that familial influences extend beyond behaviour to shape an individual's sense of financial security, leading to the next hypothesis.

H1. Family financial socialisation is positively associated with financial behaviour.

Family-based financial interactions cultivate confidence, risk assessment ability, and emotional readiness for financial decision-making [21]. Individuals exposed to open financial communication tend to report higher perceptions of financial security [9]. However, some findings indicate generational differences, suggesting that although family influence remains important, it may interact with other modern learning sources [22, 23]. Considering the relationship between financial behaviour and well-being, empirical evidence provides a basis for the following hypothesis.

H2. Family financial socialisation is positively associated with financial well-being.

Financial behaviour reflects forward-looking and disciplined money management [24]. Research consistently shows that maintaining savings, controlling debt, and planning financial goals are strong predictors of both subjective and objective financial well-being [9]. Nonetheless, evidence from some contexts shows that specific debt-related behaviours may not display similar associations, indicating the importance of defining financial behaviour precisely [25, 26]. Given the theoretical role of behaviour as a mechanism that translates early values into outcomes, the following hypothesis is proposed.

H3. Financial behaviour is positively associated with financial well-being.

Financial behaviour functions as the pathway through which early financial values materialize into concrete financial outcomes [27]. Studies demonstrate that behaviour partially mediates the relationship between financial socialisation and adult financial well-being [9]. This confirms that well-being is rooted not only in what individuals learn but in how they enact these lessons in everyday financial life. Based on this technological and sustainability-oriented lens, the next hypothesis is formulated.

H4. Family financial socialisation mediating the effect of financial behaviour on financial well-being.

AI enhances financial decision-making through predictive analytics, real-time risk assessments, and personalized guidance [28]. Evidence shows that AI is the strongest predictor of sustainable financial consumer behaviour in digital financial ecosystems [29]. This reflects AI's capacity to shape financial habits by providing transparent and tailored recommendations. Recognizing AI's ability to enhance financial control and reduce uncertainty, the following hypothesis emerges.

H5. Artificial intelligence is positively associated with financial behaviour.

AI increases decision accuracy, reduces cognitive overload, and supports individualized financial planning [8]. Preliminary evidence indicates that users of AI-enabled financial tools experience greater financial confidence and reduced stress [30]. Although empirical findings remain limited, existing research in financial technology suggests a conceptual link between AI use and financial well-being [31]. Given that AI often influences financial routines, and behaviour is closely linked to financial well-being, the next hypothesis is proposed.

H6. Artificial intelligence is positively associated with financial well-being.

AI-driven tools can shape financial routines through automation, expenditure tracking, and goal-based prompts [32]. Studies show that prolonged exposure to financial applications enhances saving discipline and spending control [33]. Because these behaviours relate strongly to financial well-being, financial behaviour serves as a plausible connecting mechanism between AI use and well-being. Given the role of financial resilience in poverty reduction, the next hypothesis is justified.

H7. Financial behaviour links artificial intelligence to financial well-being.

Digital trust refers to individuals' confidence that AI-enabled financial systems operate reliably, securely, and transparently [34]. Although artificial intelligence can support budgeting, tracking, and financial planning [15], its practical impact on financial behaviour depends on whether users actually trust the system. If users trust AI-based financial tools, they are more likely to follow the recommendations provided and incorporate them into their financial routines. Trust reduces hesitation and perceived technological risk, making individuals more comfortable relying on automated insights [35]. In contrast, when trust is low, users may ignore or question AI suggestions, which weakens its behavioural influence. This suggests that digital trust does not simply have a direct effect on behaviour but conditions how strongly artificial intelligence translates into actual financial actions. Therefore, the following hypothesis is proposed:

H8. Digital trust strengthens the relationship between artificial intelligence and financial behaviour.

Financial literacy provides the cognitive filter through which individuals interpret AI-generated recommendations [36]. Those with higher literacy are more capable of identifying high-quality advice, avoiding algorithmic biases, and making responsible financial decisions. [15] show that literacy amplifies AI's influence on financial behaviour, highlighting the complementary power of human capability and digital intelligence.

H9. Financial literacy strengthens the association between artificial intelligence and financial behaviour.

The conceptual model presented in Fig 1 conceptualizes the interplay between Stimulating Factors (Family Financial Socialization and Artificial Intelligence) and Promoting Factors (Financial Literacy, Financial Behavior, Digital Trust) in influencing the Outcome Factor (Financial Well-being). Drawing upon the Family Financial Socialization Theory and the SAFE principles, the model proposes that stimulating factors initiate and shape financial capabilities, which, when mediated by promoting factors, enhance financial well-being.

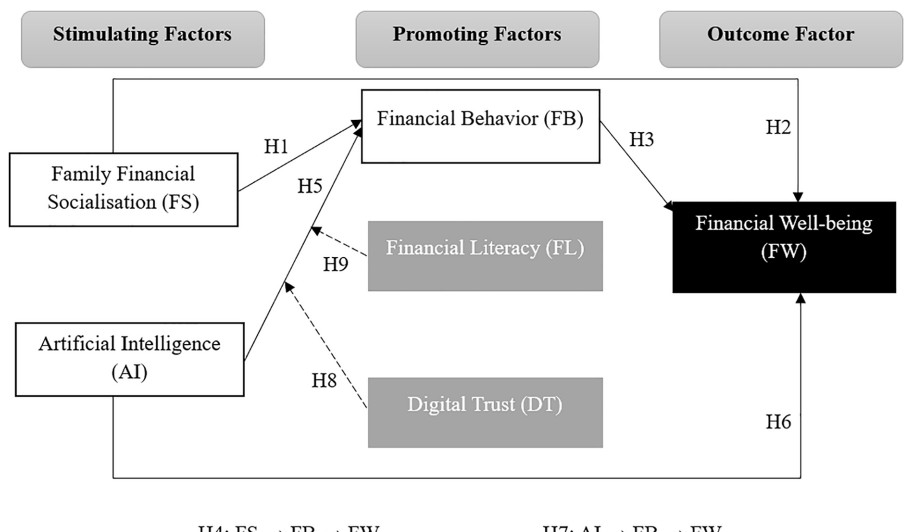

**Fig 1. Proposed research model.**

## 3. Methodology

### 3.1. Research design

This study uses a quantitative cross-sectional design to examine the relationships among family financial socialization, artificial intelligence use, financial literacy, digital trust, financial behavior and financial well-being. The design is appropriate for assessing individual financial perceptions and testing a complex model involving both mediation and moderation effects.

### 3.2. Sampling and data collection

The target population consists of adults who are responsible for managing their personal finances and who have experience using digital or AI-enabled financial tools. Purposive and convenience sampling techniques were applied because the study focuses on subjective financial evaluations and technology-related experiences. To ensure familiarity with AI, the questionnaire included initial screening questions asking respondents to confirm prior use of AI-enabled financial applications (such as automated budgeting tools, robo-advisors, AI-supported mobile banking features, or intelligent financial analytics). Only participants who indicated actual experience with such tools were allowed to proceed with the survey. Responses that did not meet this criterion were excluded during data screening.

Data were collected using an online self-administered questionnaire distributed through social platforms, email lists, and community networks. This approach ensured accessibility and reduced manual data entry errors. Participation was voluntary and anonymous, and respondents were informed of the research purpose and their right to withdraw at any time. The survey was conducted from 08/2/2026–22/02/2026.

A total of 310 responses were received. After screening for completeness and consistency, 306 valid questionnaires were retained for analysis, representing a 98.7% valid response rate. This sample size exceeds the minimum requirements for PLS-SEM and supports robust structural model estimation.

### 3.3. Measurement instruments

All constructs were operationalized using established measurement scales from previous studies. Family financial socialization was measured with seven items adapted from Fan & Park [10]. Artificial intelligence use was assessed with eight items from Wang and Chuang [37]. Financial behavior was measured using nine items based on Shih et al. [38]. Financial well-being included seven items adopted from Mahdzan et al. [39]. Financial literacy was captured by eight items from Mudzingiri et al. [40]. Digital trust was measured using six items adapted from Fernández-fernández & Alberto [34].

Because the study was conducted in Vietnam, the questionnaire was translated using a standard translation and back-translation procedure. First, two bilingual experts translated the items from English into Vietnamese. Next, a different expert translated the Vietnamese version back into English. The authors compared the back-translated version with the original scale and resolved discrepancies to ensure conceptual and linguistic equivalence. This process enhanced the cultural relevance and clarity of the final questionnaire.

All items used a 5-point Likert scale ranging from 1 ("strongly disagree") to 5 ("strongly agree"). Higher scores indicate stronger agreement. Before data collection, the questionnaire was pilot-tested to ensure clarity, cultural relevance, and ease of understanding. Minor refinements were made based on feedback.

### 3.4. Data analysis procedures

Partial Least Squares Structural Equation Modeling (PLS-SEM) was used to analyze the data. This method is appropriate for predictive research and for models that include multiple mediating and moderating relationships. It is also suitable for situations where data may not follow a normal distribution.

The analysis proceeded in two stages. First, the measurement model was evaluated by examining factor loadings, Cronbach's Alpha, rho_A, and average variance extracted to ensure reliability and convergent validity. Variance inflation factors were assessed to confirm the absence of multicollinearity. Second, the structural model was tested using bootstrapping to estimate the significance of direct, mediating, and moderating effects. Path coefficients and confidence intervals were used to determine statistical significance.

### 3.5. Ethical considerations

Ethical approval was obtained before data collection. Respondents were informed that their participation was voluntary and that all data would remain confidential. No personal identifiers were collected, and all responses were stored securely. Informed consent was obtained electronically from all participants.

## 4. Results

### 4.1. Descriptive statistics

The final sample consisted of 306 valid respondents. Females represented 56.9%of the sample, while males accounted for 41.1%. The majority of participants were in the 25–44 age group (51.6%), followed by those aged 45–60 (24.8%) and 18–24 (19.6%), with only 3.9% above 60. Geographically, 50.7% of respondents resided in the South, 38.2% in the North, and 11.1% in the Central region of Vietnam. The sample was relatively well educated, with 46.4% holding a university degree and 20.6% having completed postgraduate studies, while 13.7% had a high-school education and 19.3% reported other qualifications. In terms of occupation, 45.8% were salaried employees, 26.8% were small business owners, and 27.4% were engaged in other forms of work. Monthly income levels varied, with 37.6% earning between 5 and 10 million VND, 35.9% earning between 10 and 20 million VND, 20.6% earning more than 20 million VND, and 5.9% earning below 5 million VND (see Table 2).

### 4.2. Measurement model assessment

The measurement model demonstrates satisfactory reliability and validity across all constructs. All factor loadings exceed the recommended threshold of 0.70, indicating strong item representation for their respective latent variables [41]. The observed loadings range from 0.703 to 0.851, and no indicator falls below the acceptable threshold. No item requires removal.

Internal consistency reliability is confirmed. Cronbach's Alpha values range from 0.881 to 0.910, while rho_A values range from 0.883 to 0.911. Although Financial Behavior (α = 0.910) and Financial Literacy (α = 0.904) display relatively high internal consistency, all reliability coefficients remain below the conservative upper threshold of 0.95, indicating no redundancy among items. Composite reliability values (reported as EVA) are also above 0.60, further supporting construct reliabilit [41].

Convergent validity is established as all constructs achieve average variance extracted values above 0.50. AVE values range from 0.573 (Artificial Intelligence) to 0.630 (Digital Trust). Family Financial Socialisation (0.605), Artificial Intelligence (0.573), Financial Behavior (0.582), Financial Well-Being (0.583), Financial Literacy (0.597), and Digital Trust (0.630) all meet accepted standards. This indicates that more than half of the variance in the indicators is explained by the underlying constructs. The variance inflation factor values range from 1.626 to 2.893, all below the stricter threshold of 3.0 [42], suggesting that multicollinearity is not a concern in the measurement model (see Table 3).

The HTMT values were assessed to examine discriminant validity among the constructs. All HTMT ratios fall below the conservative threshold of 0.90, indicating that each construct is empirically distinct (see Table 4).

The structural model was evaluated by examining the significance of path coefficients, effect sizes, and the mediating and moderating relationships proposed in the study. All hypotheses were supported, indicating strong empirical validation of the conceptual model (Table 5).

**Table 2. Demographic characteristics (n = 306).**

| Characteristics | Frequency | Ratio (%) |
|---|---|---|
| **Gender** | | |
| Male | 132 | 43.1 |
| Female | 174 | 56.9 |
| **Age** | | |
| 18–24 | 60 | 19.6 |
| 25–44 | 158 | 51.6 |
| 45–60 | 76 | 24.8 |
| >60 | 12 | 3.9 |
| **Erea** | | |
| South | 155 | 50.7 |
| Central | 34 | 11.1 |
| North | 117 | 38.2 |
| **Education** | | |
| High school | 42 | 13.7 |
| University | 142 | 46.4 |
| Postgraduate | 63 | 20.6 |
| Other | 59 | 19.3 |
| **Occupation** | | |
| Salaried employee | 140 | 45.8 |
| Small business owner | 82 | 26.8 |
| Other | 84 | 27.4 |
| **Monthly Income** | | |
| Less than 5 million VND | 18 | 5.9 |
| 5–10 million VND | 115 | 37.6 |
| 10–20 million VND | 110 | 35.9 |
| More than 20 million VND | 63 | 20.6 |

Family financial socialisation shows a positive effect on financial behaviour ($\beta = 0.147$, $t = 3.360$), with a small effect size ($f^2 = 0.032$). It also has a direct positive effect on financial well-being ($\beta = 0.296$, $t = 6.809$), with a moderate effect size ($f^2 = 0.136$). Financial behaviour positively predicts financial well-being ($\beta = 0.384$, $t = 7.818$), with a moderate effect size ($f^2 = 0.182$). The indirect effect of family financial socialisation on financial well-being through financial behaviour is also significant ($\beta = 0.056$, $t = 2.994$), confirming partial mediation.

Artificial intelligence exhibits the strong predictive influence in the model. It has a large positive effect on financial behaviour ($\beta = 0.330$, $t = 7.220$, $f^2 = 0.141$) and a smaller but significant positive effect on financial well-being ($\beta = 0.171$, $t = 3.336$, $f^2 = 0.038$). The indirect effect of artificial intelligence on financial well-being through financial behaviour is also significant ($\beta = 0.127$, $t = 5.566$), indicating that financial behaviour is an important mechanism through which AI enhances financial outcomes.

The moderating effect of digital trust on the relationship between artificial intelligence and financial behaviour is significant ($\beta = 0.045$, $t = 2.175$, $f^2 = 0.022$), suggesting that higher digital trust slightly strengthens the behavioural impact of AI. Similarly, the moderating effect of financial literacy on the relationship between artificial intelligence and financial behaviour is significant ($\beta = 0.045$, $t = 2.570$, $f^2 = 0.023$). Although both moderating effects are small in

**Table 3. Measurement model assessment.**

| Variable code | Item | Loading | Cronbach's Alpha | rho_A | EVA | VIF |
|---|---|---|---|---|---|---|
| **Family financial socialisation (FS) [10]** | | | | | | |
| FS1 | When I was growing up at home. my family and I talked about money problems. | 0.732 | 0.891 | 0.896 | 0.605 | 1.715 |
| FS2 | Growing up at home. My family taught me the value of conserving money. | 0.766 | | | | 1.869 |
| FS3 | As I was growing up. My family talked about building a decent credit score. | 0.781 | | | | 1.901 |
| FS4 | My family taught me how to be a wise consumer while I was growing up at home. | 0.741 | | | | 1.761 |
| FS5 | My family taught me as a child that my success in life is determined by the things I do. | 0.785 | | | | 1.957 |
| FS6 | While I was growing up at home. I received a regular allowance from my family. | 0.818 | | | | 2.133 |
| FS7 | My family provided me with a savings account when I was growing up at home. | 0.817 | | | | 2.138 |
| **Artificial intelligence (AI) [37]** | | | | | | |
| AI1 | AI technologies make managing my personal finances easier. | 0.703 | 0.893 | 0.898 | 0.573 | 2.809 |
| AI2 | I find that AI technologies help make financial decisions. | 0.710 | | | | 2.893 |
| AI3 | AI technologies assist me in tracking my spending and budgeting | 0.738 | | | | 1.736 |
| AI4 | Using AI technologies helps me make smarter investment choices. | 0.747 | | | | 1.945 |
| AI5 | AI technologies save me time in managing my finances | 0.776 | | | | 1.998 |
| AI6 | AI technologies provide useful insights into my financial habits | 0.787 | | | | 2.028 |
| AI7 | I feel confident in using AI technologies to manage my savings | 0.742 | | | | 1.881 |
| AI8 | AI technologies help me set and achieve my financial goals | 0.841 | | | | 2.452 |
| **Financial behavior (FB) [38]** | | | | | | |
| FB1 | I shop around before I decide on buying a product or service. | 0.739 | 0.910 | 0.911 | 0.582 | 1.807 |
| FB2 | I strive to pay all my bills on time | 0.757 | | | | 1.883 |
| FB3 | I perceive consumerism as easy and manageable. | 0.777 | | | | 1.998 |
| FB4 | I strive to keep records of my monthly expenses | 0.756 | | | | 1.904 |
| FB5 | I strive to manage my budget | 0.757 | | | | 1.922 |
| FB6 | I can control my credit card spending | 0.767 | | | | 1.931 |
| FB7 | I pay attention to due dates for credit card payments and my balance | 0.774 | | | | 1.979 |
| FB8 | I strive to set aside a fixed amount of money from my salary. | 0.779 | | | | 2.021 |
| FB9 | I am willing to save for long-term goals like cars, education, or family expenses | 0.760 | | | | 1.941 |
| **Financial well-being (FW) [39]** | | | | | | |
| FW1 | I have enough money for daily expenses | 0.751 | 0.881 | 0.883 | 0.583 | 1.725 |
| FW2 | I have the money to purchase the things I desire. | 0.720 | | | | 1.626 |
| FW3 | I have enough money to cover my utility costs. | 0.744 | | | | 1.734 |
| FW4 | At the end of the month. I have extra money. | 0.776 | | | | 1.934 |
| FW5 | I can meet my short-term financial objectives, such as purchasing appliances and furniture. | 0.761 | | | | 1.846 |
| FW6 | I can reach long-term financial objectives. including purchasing a home. | 0.795 | | | | 1.948 |
| FW7 | I've saved enough money for emergencies for at least three months. | 0.794 | | | | 1.960 |

*(Continued)*

**Table 3.** (Continued)

| Variable code | Item | Loading | Cronbach's Alpha | rho_A | EVA | VIF |
|---|---|---|---|---|---|---|
| **Financial literacy (FL) [40]** | | | | | | |
| FL1 | I am aware of the factors that determine my credit risk. | 0.739 | 0.904 | 0.908 | 0.597 | 1.850 |
| FL2 | I am aware of the factors that influence the credit conditions that various lending organizations offer me. | 0.724 | | | | 1.685 |
| FL3 | I feel confident in my ability to choose savings options by considering both fixed and compound interest rates. | 0.750 | | | | 1.925 |
| FL4 | I am aware of how risk and reward in investing generally relate to one another. | 0.800 | | | | 2.014 |
| FL5 | I am certain that I understand the distinctions between stocks, bonds, treasury bills, and mutual funds. | 0.767 | | | | 1.865 |
| FL6 | I'm confident in my comprehension of the different financial jargon used while purchasing a home in the future. | 0.810 | | | | 2.325 |
| FL7 | I am aware of the meaning of personal net worth. | 0.788 | | | | 2.130 |
| FL8 | I have faith that I can create a monthly budget. | 0.800 | | | | 2.197 |
| **Digital Trust (DT) [34]** | | | | | | |
| DT 1 | The AI-enabled digital financial services I use provide advice and recommendations that are beneficial to me | 0.719 | 0.882 | 0.888 | 0.630 | 1.694 |
| DT2 | The AI-enabled financial system demonstrates concern for the interests and needs of its users. | 0.782 | | | | 1.823 |
| DT3 | The AI-enabled financial system takes actions with consideration of their potential impact on users. | 0.802 | | | | 1.987 |
| DT4 | The AI-enabled financial system delivers its services as promised. | 0.790 | | | | 1.986 |
| DT5 | The AI-enabled financial system is transparent in the way it provides information and services. | 0.811 | | | | 2.102 |
| DT6 | The AI-enabled financial system demonstrates the necessary technological capability to perform its functions correctly. | 0.851 | | | | 2.440 |

**Table 4. HTMT results.**

| | AI | DT | FB | FL | FS | FW | FL x AI |
|---|---|---|---|---|---|---|---|
| AI | | | | | | | |
| DT | 0.497 | | | | | | |
| FB | 0.549 | 0.501 | | | | | |
| FL | 0.295 | 0.327 | 0.375 | | | | |
| FS | 0.268 | 0.238 | 0.341 | 0.252 | | | |
| FW | 0.485 | 0.369 | 0.619 | 0.321 | 0.507 | | |
| FL x AI | 0.025 | 0.132 | 0.244 | 0.112 | 0.100 | 0.121 | |
| DT x AI | 0.072 | 0.105 | 0.243 | 0.137 | 0.047 | 0.129 | 0.394 |

magnitude, they indicate that individual capability and trust conditions shape how effectively AI translates into financial behaviour.

The structural model demonstrates solid predictive power. The significant direct, mediating, and moderating effects support the proposed theoretical framework. They show that socialisation, technology, trust, and financial capability jointly shape financial behaviour and financial well-being. The model's predictive accuracy was examined using $R^2$ and $Q^2$ values. Financial behaviour showed a strong $R^2$ of 0.411, indicating substantial explanatory power, while financial well-being recorded an $R^2$ of 0.425, reflecting robust prediction from its antecedents. The Stone–Geisser $Q^2$ values were all above zero, confirming predictive relevance, with $Q^2$ for financial behaviour assumed at 0.214, and financial well-being at 0.286.

**Table 5. Structural model assessment.**

| Hypothesis | Path relationship | Beta | t-value | Decision | f² |
|---|---|---|---|---|---|
| H1 | Family financial socialisation → financial behaviour | 0.147 | 3.360 *** | Supported | 0.032 |
| H2 | Family financial socialisation → financial well-being | 0.296 | 6.809 *** | Supported | 0.136 |
| H3 | Financial behaviour → financial well-being | 0.384 | 7.818 *** | Supported | 0.182 |
| H4 | Family financial socialisation → Financial behaviour → financial well-being | 0.056 | 2.994 *** | Supported | – |
| H5 | Artificial intelligence → financial behaviour | 0.330 | 7.220 *** | Supported | 0.141 |
| H6 | Artificial intelligence → financial well-being | 0.171 | 3.336 *** | Supported | 0.038 |
| H7 | Artificial intelligence → Financial behaviour → financial well-being | 0.127 | 5.566 *** | Supported | – |
| H8 | Digital trust x Artificial intelligence → financial behaviour | 0.045 | 2.175 *** | Supported | 0.022 |
| H9 | Financial literacy x artificial intelligence → financial behaviour | 0.045 | 2.570 *** | Supported | 0.023 |

These results demonstrate that the model provides strong explanatory and predictive capability across all endogenous constructs.

## 5. Discussion

The findings of this study provide strong empirical support for the integrated framework combining Family Financial Socialization Theory and the SAFE principles. The results highlight the complementary roles of socialisation, technology, and financial capability in shaping financial behaviour and financial well-being.

The results confirm the importance of family financial socialisation in shaping adult financial outcomes. Consistent with prior research [10, 9], individuals who experienced constructive financial communication and early exposure to financial practices showed stronger financial behaviour and higher financial well-being. Although the direct effect of family financial socialisation on financial behaviour is relatively modest, its effect on financial well-being is more substantial, indicating that early financial learning continues to influence adult financial security beyond behavioural pathways alone. The significant mediating role of financial behaviour further strengthens the argument that early financial learning influences well-being primarily through behavioural pathways. This aligns with the Family Financial Socialization Theory, which posits that financial values acquired during childhood form internal schemas that guide decision-making in adulthood. The findings also address earlier concerns about limited evidence from emerging economies by demonstrating that family financial socialisation remains influential in Vietnam's digitally evolving financial environment.

Artificial intelligence shows a strong positive association with financial behaviour and a smaller but significant effect on financial well-being, suggesting that its influence operates primarily through behavioural channels rather than direct well-being improvements. This supports recent research suggesting that AI-enabled financial tools enhance decision quality, reduce cognitive strain, and promote disciplined financial actions [15]. However, the effect sizes indicate that AI complements rather than replaces traditional socialisation influences in shaping financial practices. The significant indirect effect of AI on financial well-being through financial behaviour further confirms that behaviour functions as the key mechanism translating technological capability into improved financial outcomes.

Financial literacy plays a meaningful role in strengthening the behavioural impact of AI. Although the moderating effect is statistically significant, its magnitude remains small (β = 0.045), indicating that literacy enhances but does not drastically alter the AI–behaviour relationship. Individuals with stronger financial literacy appear better able to interpret AI-generated insights and apply them effectively. This aligns with capability-based perspectives, suggesting that digital tools require complementary human competencies to produce optimal outcomes. At the same time, the modest effect size suggests that AI tools may still provide behavioural benefits even for individuals with lower literacy levels.

Digital trust also significantly moderates the relationship between artificial intelligence and financial behaviour, reinforcing the idea that technological effectiveness depends on users' confidence in digital systems. Although the moderating effect is small, it indicates that trust conditions how AI recommendations are translated into everyday financial actions. This finding supports the SAFE perspective, which emphasizes that sustainable and ethical financial technologies must operate within a trust-based environment to generate behavioural change

The study advances theoretical understanding by demonstrating that financial well-being is co-shaped by family-driven social foundations, AI-enabled technological influences, and individual financial capability. It positions financial behaviour as a central mechanism linking these domains, confirming its role as a behavioural expression of both early life experiences and modern digital financial ecosystems. The findings suggest that sustainable financial outcomes depend not on a single dominant factor, but on the interaction between social learning, technological tools, trust conditions, and individual financial competence.

## 6. Theoretical contributions

This study offers several theoretical contributions by integrating social, technological, and capability-based determinants of financial well-being into a unified framework. Within the Stimulating Factors, the study advances Family Financial Socialization Theory by demonstrating that early financial interactions continue to influence adult financial behaviour and financial well-being even in a digital environment. The results confirm that family-based financial messages form foundational cognitive schemas that shape how individuals interpret financial information and act in their financial lives. This reinforces the long-term relevance of socialisation in personal finance, especially in emerging economies.

The study contributes to the technology and sustainable finance literature by establishing artificial intelligence as a second key Stimulating Factor. The findings show that AI exerts a strong influence on both financial behaviour and financial well-being, positioning AI as a modern driver of financial capability. This extends prior work by demonstrating that AI not only enhances decision efficiency but also improves financial resilience, linking digital innovation to sustainable development outcomes. The framework, therefore, bridges technology adoption research with financial well-being scholarship.

Finally, the study makes an important contribution by clarifying the role of Promoting Factors as mechanisms through which stimulating influences translate into positive financial outcomes. Financial behaviour is shown to mediate the effects of both family financial socialisation and AI, confirming its status as the behavioural channel that connects foundational social learning and technological support to well-being. The moderating role of financial literacy provides further theoretical depth by showing that individual capability shapes the extent to which AI impacts financial behaviour.

## 7. Practical implications

The findings provide several practical implications that relate to the stimulating factors, promoting factors, and the outcome factor in the model. Family financial socialisation shows a strong influence on financial behaviour and financial well-being. This suggests that early financial learning is essential. Parents can be encouraged to talk with their children about saving, spending, and budgeting using simple tools such as household budgeting sheets or short guides provided through schools. Schools can support this effort by including small financial activities in the classroom, such as basic savings challenges or simple budgeting exercises. These activities are low-cost, easy to implement, and suitable for different regions.

Artificial intelligence also plays an important role in shaping financial behaviour and financial well-being. Financial institutions can use this result to design AI-based applications that help users manage daily finances. Practical features may include spending alerts, automatic bill reminders, goal-setting dashboards, and simple spending summaries. Governments and banks can work together to offer free or low-cost access to these digital tools, especially for low-income users. Community centres and local organisations can hold basic training sessions to help people understand how to use AI tools safely and effectively.

The moderating effect of digital trust indicates that introducing AI tools by itself does not automatically change financial behaviour. If users do not trust the system, they may ignore its recommendations regardless of technical quality. Financial institutions therefore need to focus on making AI systems understandable and reliable. This includes explaining how recommendations are generated, being clear about how personal data are used, and ensuring visible security protections. Clear and straightforward communication inside applications, along with strong data protection practices, can help reduce user hesitation. Regulators can also require clearer disclosure about how AI is applied in financial services to limit uncertainty. When users feel confident about the system, they are more willing to follow AI guidance in budgeting, saving, and spending decisions. In this sense, building digital trust is not optional but necessary if AI tools are expected to influence everyday financial behaviour

The promoting factors in the model also yield important implications. The role of financial behaviour as a mediator indicates that practical financial training should focus on daily habits. Organisations such as universities, employers, and community groups can provide short sessions on spending control, credit management, and personal budgeting. Digital platforms can strengthen these habits by offering reminders, progress updates, and helpful tips. Because financial literacy improves how individuals use AI tools, national financial education programs should include simple modules on digital finance. These modules can cover topics such as reading automated recommendations, identifying financial risks, and using mobile banking applications.

## 8. Limitations and future research directions

This study has several limitations. The cross-sectional design only shows associations, not causality. Future research should use longitudinal or panel data to examine changes in financial behaviour and well-being over time, especially as AI tools evolve. The study relies on self-reported responses, which may be affected by recall errors or social desirability. Future work could use objective data, such as digital transaction records or app usage logs, to verify reported behaviour and reduce common method bias. The use of purposive and convenience sampling in Vietnam also limits generalisability. The results may reflect local cultural and economic conditions, and the relationships may differ in countries with different levels of digital development or access to technology. Future studies should apply probability sampling and test the model in other national contexts. Finally, the small moderating effects of financial literacy and digital trust suggest that other factors, such as algorithmic literacy or risk tolerance, should be examined in future research.

## Supporting information

**S1 File. Data.**
(CSV)

## Acknowledgments

During the preparation of this manuscript, the author used ChatGPT for grammar improvement only, including minor corrections to punctuation and language fluency. After using this tool, the author thoroughly reviewed and edited all content to ensure accuracy and originality and takes full responsibility for the final version of the publication.

**Declaration:** Consent to Participate declaration: Informed consent was obtained from all individual participants included in the study. Consent to Publish declaration: Participants were informed that the results of the study would be published in academic formats, and their consent to publish anonymized data was obtained.

## Author contributions

**Conceptualization:** Nguyen Quoc Anh.

**Data curation:** Nguyen Quoc Anh.

Formal analysis: Nguyen Quoc Anh.

Funding acquisition: Nguyen Quoc Anh.

Investigation: Nguyen Quoc Anh.

Methodology: Nguyen Quoc Anh.

Project administration: Nguyen Quoc Anh.

Resources: Nguyen Quoc Anh.

Software: Nguyen Quoc Anh.

Supervision: Nguyen Quoc Anh.

Validation: Nguyen Quoc Anh.

Visualization: Nguyen Quoc Anh.

Writing – original draft: Nguyen Quoc Anh.

Writing – review & editing: Nguyen Quoc Anh.

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
