## [Decision Letter · Decision Letter 0]

6 Feb 2026

Dear Dr. Anh,

Thank you for submitting your manuscript to PLOS ONE. After careful consideration, we feel that it has merit but does not fully meet PLOS ONE’s publication criteria as it currently stands. Therefore, we invite you to submit a revised version of the manuscript that addresses the points raised during the review process.

Dear authors,

Following the feedback from the two reviewers, the manuscript requires a major revision.

I propose to systematically address each reviewer’s concerns. Looking forward to your input.

Best regards

We look forward to receiving your revised manuscript.

Kind regards,

Manuel Herrador, Ph.D.

Academic Editor

PLOS One

“This research is funded (supported) by University of Economics Ho Chi Minh City, Vietnam (UEH).”

“This research is funded (supported) by University of Economics Ho Chi Minh City, Vietnam (UEH).”

“This research is funded (supported) by University of Economics Ho Chi Minh City, Vietnam (UEH).”

Additional Editor Comments:

Dear authors,

Following the feedback from the two reviewers, the manuscript requires a major revision.

I propose to systematically address each reviewer’s concerns. Looking forward to your input.

Best regards

Reviewers' comments:

Reviewer's Responses to Questions

**Comments to the Author**

1. Is the manuscript technically sound, and do the data support the conclusions?

Reviewer #1: No

Reviewer #2: Yes

2. Has the statistical analysis been performed appropriately and rigorously?

Reviewer #1: Yes

Reviewer #2: Yes

3. Have the authors made all data underlying the findings in their manuscript fully available?

Reviewer #1: No

Reviewer #2: Yes

4. Is the manuscript presented in an intelligible fashion and written in standard English?

Reviewer #1: Yes

Reviewer #2: Yes

Reviewer #1: 1. SDG 1 is Conceptually Misapplied

SDG 1 is operationalised through income poverty, multidimensional poverty (MPI), access to basic services, social protection, and vulnerability to shocks. None of the constructs used in H1–H4 measure or validly proxy any of these. The model instead captures:

• perceptions,

• participation/engagement,

• behavioural or attitudinal change,

• technology interaction.

These relate to developmental processes, not poverty reduction. The paper conflates inclusion or engagement with poverty alleviation, an error well established in poverty scholarship (Sen; Alkire & Foster; Ravallion).

2. No Poverty Variable, No Poverty Proxy

There is:

• no income/consumption data,

• no MPI dimensions,

• no assets or services indicators,

• no vulnerability/resilience measure.

Yet the hypotheses are framed as supporting SDG 1. This is methodologically indefensible.

3. Post-hoc SDG Framing

The SDG 1 linkage appears retrofitted rather than theory-driven. Proper alignment would require starting with the SDG indicators and showing how variables map to them. That is absent here.

4. Missing Poverty Literature

There is no engagement with the literature on capability, multidimensional poverty, or vulnerability. As a result, the SDG claim is rhetorical, not scholarly.

5. What the Model Actually Relates To

Up to H4, the model more plausibly relates to:

• social inclusion,

• engagement,

• innovation/technology use,

• community processes

—closer to SDGs 9, 10, 11, or 17, but not SDG 1.

6. Major Methodological Gap: AI Familiarity of Respondents

The paper does not explain how subjects were selected to ensure familiarity with AI, which is critical given that the model depends on perceptions and interactions with AI systems. Without establishing baseline familiarity or exposure, responses risk reflecting misunderstanding rather than informed evaluation, weakening the validity of all hypotheses.

Reviewer #2: The manuscript addresses a highly significant issue by linking individual financial well-being to the global effort to eradicate poverty, specifically Sustainable Development Goal 1 (SDG1). The sources indicate that financial well-being is fundamental to social inclusion and economic resilience, making it a pertinent topic for social work practitioners and policymakers. The work contributes to existing knowledge by integrating social, technological, and capability-based elements into a unified framework, moving beyond fragmented studies that often examine these factors in isolation. Furthermore, it provides valuable empirical evidence from Vietnam, an emerging economy where such research is currently scarce.

The aims and objectives of the study are clearly stated and well-aligned with the overall research design. The manuscript seeks to explain how financial socialization, technology (specifically Artificial Intelligence), and financial capability jointly shape financial behavior and well-being. The hypotheses (H1–H9) are logically derived from the literature and directly correspond to the proposed research model.

The article is firmly grounded in interdisciplinary frameworks, utilizing the Family Financial Socialization Theory, the SDGs framework, and the SAFE Principles (Sustainability, Accountability, Fairness, and Ethics). The sources demonstrate critical engagement with existing literature by identifying the limitations of these theories—such as their failure to account for digital capability or algorithmic trust—and proposing an integrated model to address these gaps.

The research methodology is extensively described and appears appropriate for the study’s predictive nature. Ethical considerations were thoroughly addressed; the study received approval from the University of Economics Ho Chi Minh City Ethics Committee, and informed consent was obtained from all participants.

The findings are well-organized and presented through detailed statistical assessments of both the measurement and structural models.

The manuscript offers meaningful practical and policy implications.

The manuscript is well-structured and flows logically from theoretical foundations to empirical results. The author acknowledges the use of ChatGPT for grammar improvement and language fluency, while taking full responsibility for the accuracy and originality of the content. Tables and figures, such as the proposed research model and demographic characteristics, are used appropriately to clarify the data. References appear to be formatted correctly according to standard academic practices.

To further improve the manuscript, I offer the following specific suggestions:

1. Since the study uses a cross-sectional design, the author should explicitly state that causal inferences cannot be definitively drawn and suggest longitudinal research for future studies.

2. The reliance on non-probability sampling limits the generalizability of the findings; the author should discuss how this might affect the results in different cultural or economic contexts.

3. While self-reported data is standard, future iterations could benefit from including objective behavioral records, such as digital transaction logs, to validate participants' perceptions.

4. Given the small moderating effect of financial literacy, the author could expand the discussion on other factors, such as digital trust or algorithmic literacy, that might influence AI-driven financial behavior.

**Do you want your identity to be public for this peer review?** For information about this choice, including consent withdrawal, please see our Privacy Policy

Reviewer #1: No

Reviewer #2: No

---

## [Author Response · Author response to Decision Letter 1]

26 Feb 2026

I would like to express my sincere gratitude to the reviewers for their insightful and constructive comments. In the revised manuscript, all modifications made in response to the reviewers’ suggestions.

Review #1

Review Comments to the Author

1. SDG 1 is Conceptually Misapplied

SDG 1 is operationalised through income poverty, multidimensional poverty (MPI), access to basic services, social protection, and vulnerability to shocks. None of the constructs used in H1–H4 measure or validly proxy any of these. The model instead captures:

• perceptions,

• participation/engagement,

• behavioural or attitudinal change,

• technology interaction.

These relate to developmental processes, not poverty reduction. The paper conflates inclusion or engagement with poverty alleviation, an error well established in poverty scholarship (Sen; Alkire & Foster; Ravallion).

Responses

Thank you for raising this point. I agree that SDG1 is defined using objective poverty indicators such as income poverty and vulnerability to shocks, and these are not directly measured in my model. I have revised the manuscript accordingly. SDG1 is no longer treated as a direct outcome variable. The model now focuses on financial well-being at the individual level. The study does not claim to measure poverty reduction. Instead, it examines how socialisation and AI-related factors influence financial behaviour and financial well-being. Any reference to SDG1 is now limited to background context, not empirical testing.

Review Comments to the Author

2. No Poverty Variable, No Poverty Proxy

There is:

• no income/consumption data,

• no MPI dimensions,

• no assets or services indicators,

• no vulnerability/resilience measure.

Yet the hypotheses are framed as supporting SDG 1. This is methodologically indefensible.

Responses

I have removed all statements framing the hypotheses as empirically supporting SDG1. The revised manuscript limits the analysis to financial behaviour and financial well-being at the individual level. The study no longer presents its findings as evidence of poverty reduction. Any mention of SDG1 is now positioned only as broader policy context, not as a tested outcome.

Review Comments to the Author

3. Post-hoc SDG Framing

The SDG 1 linkage appears retrofitted rather than theory-driven. Proper alignment would require starting with the SDG indicators and showing how variables map to them. That is absent here.

Responses

I have removed the SDG1 framing from the core theoretical model. The revised manuscript no longer positions the hypotheses as being derived from SDG indicators, nor does it claim direct alignment with SDG1 measurement dimensions. The study is framed around Family Financial Socialization Theory and the SAFE principles, focusing on financial behaviour and financial well-being as the primary constructs.

Review Comments to the Author

4. Missing Poverty Literature

There is no engagement with the literature on capability, multidimensional poverty, or vulnerability. As a result, the SDG claim is rhetorical, not scholarly.

Responses

I have removed the explicit SDG1 claim and avoided framing the study as a contribution to poverty measurement or multidimensional poverty scholarship. The revised manuscript now situates the study within financial well-being, financial behaviour, and technology adoption literature, without extending its claims to poverty capability frameworks.

Review Comments to the Author

5. What the Model Actually Relates To

Up to H4, the model more plausibly relates to:

• social inclusion,

• engagement,

• innovation/technology use,

• community processes

—closer to SDGs 9, 10, 11, or 17, but not SDG 1.

Responses

In the revised manuscript, I have removed the positioning of the model as supporting SDG1. The study is framed as focusing on financial behaviour, financial well-being, and the role of AI and capability factors at the individual level. I do not reassign the model to alternative SDGs, as the primary objective is not SDG testing but theory-driven examination of financial behaviour and well-being.

Thank you for pointing this out. The comment helped me see where the argument was overstated and where the framing needed to be tightened. I have revised the manuscript to reflect the actual scope of the study. I hope these changes address the concern and I would appreciate your reconsideration of the revised version.

Review Comments to the Author

6. Major Methodological Gap: AI Familiarity of Respondents

The paper does not explain how subjects were selected to ensure familiarity with AI, which is critical given that the model depends on perceptions and interactions with AI systems. Without establishing baseline familiarity or exposure, responses risk reflecting misunderstanding rather than informed evaluation, weakening the validity of all hypotheses.

Responses

I have clarified the sampling procedure in the revised manuscript. Screening questions were included at the beginning of the survey to confirm that respondents had prior experience using AI-enabled financial tools (e.g., AI-supported banking features, budgeting apps, or robo-advisors). Only participants who confirmed such experience were retained for analysis, and ineligible responses were excluded during data cleaning (page 9).

Review #2

Review Comments to the Author

Since the study uses a cross-sectional design, the author should explicitly state that causal inferences cannot be definitively drawn and suggest longitudinal research for future studies.

Responses

I have revised the limitations section to explicitly state that, due to the cross-sectional design, the study identifies associations rather than definitive causal relationships. I have also clearly suggested longitudinal or panel research as a direction for future studies to better examine causal dynamics over time (page 24).

Review Comments to the Author

The reliance on non-probability sampling limits the generalizability of the findings; the author should discuss how this might affect the results in different cultural or economic contexts.

Responses

I have revised the limitations section to clarify that the use of purposive and convenience sampling in Vietnam may limit generalisability. I now explicitly state that the findings may be influenced by specific cultural and economic conditions, and that the relationships observed could differ in other contexts. I have also suggested cross-country studies and probability sampling for future research (page 24).

Review Comments to the Author

While self-reported data is standard, future iterations could benefit from including objective behavioral records, such as digital transaction logs, to validate participants' perceptions.

Responses

I have revised the limitations section to acknowledge the reliance on self-reported data and its potential biases. The manuscript now explicitly states that future research should incorporate objective behavioural records, such as digital transaction logs or app usage data, to validate reported financial behaviour and strengthen measurement accuracy (page 24).

Review Comments to the Author

Given the small moderating effect of financial literacy, the author could expand the discussion on other factors, such as digital trust or algorithmic literacy, that might influence AI-driven financial behavior.

Responses

I have expanded the discussion section to address this point. The revised manuscript now acknowledges that the moderating effect of financial literacy is relatively small and discusses the potential role of digital trust in shaping AI-driven financial behaviour. I also note that future research should examine these additional capability-related variables to better understand how individuals interact with AI-based financial tools (page 9).

Following the reviewer 1, SDG1 has been removed from the empirical model, as the study does not include objective poverty indicators. To ensure consistency between the variables and the theoretical framing, the manuscript title has also been revised and the reference to SDG1 has been deleted (page 1).

---

## [Editor Report · Decision Letter 1]

3 Mar 2026

An Integrated Model of Financial Socialization, Technology, and Financial Capability in Predicting Financial Well-Being

PONE-D-25-63897R1

Dear Dr. Anh,

We’re pleased to inform you that your manuscript has been judged scientifically suitable for publication and will be formally accepted for publication once it meets all outstanding technical requirements.

Kind regards,

Manuel Herrador, Ph.D.

Academic Editor

PLOS One

Additional Editor Comments (optional):

Dear authors,

Thanks for submitting your work to PLOS ONE.

Congratulations, the reviewers have no further comments, thus, we acknowledge the value of this work at its current stage for publication.

Best regards